AMSF: attention-based multi-view slice fusion for early diagnosis of Alzheimer’s disease

Zhang Yameng 1
Peng Shaokang 2
Xue Zhihua 3
Zhao Guohua 4
Li Qing 5
Zhu Zhiyuan 6
Gao Yufei 2 yfgao@zzu.edu.cn
Kong Lingfei 1 lfkong@zzu.edu.cn
for the Alzheimer’s Disease Neuroimaging Initiative
1 Department of Pathology, Henan Provincial People’s Hospital, People’s Hospital of Zhengzhou University , Zhengzhou , China
2 School of Cyber Science and Engineering, Zhengzhou University, SongShan Laboratory , Zhengzhou , China
3 Laboratory Animal Center, Academy of Medical Sciences, Zhengzhou University , Zhengzhou , China
4 Department of Magnetic Resonance Imaging, The First Affiliated Hospital of Zhengzhou University , Zhengzhou , China
5 State Key Laboratory of Cognitive Neuroscience and Learning, Beijing Normal University , Beijing , China
6 School of Communication and Information Engineering, Chongqing University of Posts and Telecommunications , Chongqing , China
Alatas Bilal
Electronic publication date: 2023 Nov 23
Publication date: 2023
Volume: 9
Electronic Location ID: e1706
Received 2023 Aug 18; Accepted 2023 Oct 26
Copyright: © 2023 Zhang et al.
Copyright year: 2023
Copyright holder: Zhang et al.
License: This is an open access article distributed under the terms of the Creative Commons Attribution License, which permits unrestricted use, distribution, reproduction and adaptation in any medium and for any purpose provided that it is properly attributed. For attribution, the original author(s), title, publication source (PeerJ Computer Science) and either DOI or URL of the article must be cited.
License URL: https://creativecommons.org/licenses/by/4.0/

Keywords: Alzheimer’s disease, Magnetic resonance imaging, Attention mechanism, Multi-view slice fusion

Funding: Nature Science Foundation of China 62006210, 62001284, 62206252 Key Scientific and Technology Project in Henan Province of China 221100210100 Key Project of Collaborative Innovation in Nanyang 22XTCX12001 Research Foundation for Advanced Talents of Zhengzhou University 32340306 This work was supported by the Nature Science Foundation of China (62006210, 62001284, 62206252), the Key Scientific and Technology Project in Henan Province of China (221100210100), the Key Project of Collaborative Innovation in Nanyang (22XTCX12001), the Research Foundation for Advanced Talents of Zhengzhou University (32340306). The funders had no role in study design, data collection and analysis, decision to publish, or preparation of the manuscript.

==============================
Alzheimer’s disease (AD) is an irreversible neurodegenerative disease with a high prevalence in the elderly population over 65 years of age. Intervention in the early stages of AD is of great significance to alleviate the symptoms. Recent advances in deep learning have shown extreme advantages in computer-aided diagnosis of AD. However, most studies only focus on extracting features from slices in specific directions or whole brain images, ignoring the complementarity between features from different angles. To overcome the above problem, attention-based multi-view slice fusion (AMSF) is proposed for accurate early diagnosis of AD. It adopts the fusion of three-dimensional (3D) global features with multi-view 2D slice features by using an attention mechanism to guide the fusion of slice features for each view, to generate a comprehensive representation of the MRI images for classification. The experiments on the public dataset demonstrate that AMSF achieves 94.3% accuracy with 1.6–7.1% higher than other previous promising methods. It indicates that the better solution for AD early diagnosis depends not only on the large scale of the dataset but also on the organic combination of feature construction strategy and deep neural networks.

Introduction

Alzheimer’s disease (AD) is a neurodegenerative disease with a high prevalence in people over 65 years of age (Reiman et al., 2012). Previous studies have shown that the structural changes in the brain caused by AD can be traced back 20 years before the onset of symptoms in patients (Barthelemy et al., 2020). In the early stage of AD, patients may not notice any significant changes in their brain structure or activity, but some difficulties with memory recall or retention may appear. As AD progresses, it leads to the formation of brain tissue lesions that impair and ultimately destroy neurons responsible for various cognitive functions (Zarei et al., 2013), including deterioration of memory and thinking skills, as well as a decline in physical abilities and independence. Patients with AD may present symptoms such as memory loss, cognitive impairment, language difficulties, and reduced mobility (Gaugler et al., 2022).

The global community is currently confronted with a significant demographic predicament characterized by a rapid expansion in the population of older individuals. As per the United Nations, the proportion of individuals aged 65 years and above in the overall global population is projected to reach 9.7% by 2022, and further escalate to 16.4% by 2050 (United Nations, 2023). This unprecedented surge in ageing demographics presents formidable challenges for healthcare systems, given the heightened vulnerability of older adults to chronic and degenerative ailments. Among these conditions, AD stands out as a highly prevalent and debilitating disorder that profoundly impacts the cognitive and functional capacities of countless individuals across the globe.

Mild cognitive impairment (MCI) represents a pivotal transitional phase between normal ageing and dementia, characterized by a discernible cognitive decline that does not significantly impede daily functioning. MCI assumes a critical role as an early intervention window, presenting a valuable opportunity to mitigate or forestall subsequent cognitive deterioration (Wee et al., 2012). Extensive research has established that individuals diagnosed with MCI face a heightened susceptibility to developing AD, with an annual conversion rate ranging from 10% to 15% (Roberts & Knopman, 2013). Consequently, the implementation of timely and efficacious medical interventions during the MCI stage holds the potential to safeguard neural cells against further impairment and delay the onset of AD pathology, thereby contributing to a reduction in the mortality associated with this incurable affliction (Odusami, Maskeliunas & Damasevicius, 2022).

Neuroimaging serves as a valuable and indispensable tool in the clinical diagnosis of AD, enabling the quantification of structural and functional alterations within the brain that accompany disease progression. Among the diverse array of neuroimaging modalities available, magnetic resonance imaging (MRI) has garnered considerable attention owing to its high spatial resolution and non-invasive characteristics. Through MRI, intricate details regarding brain volume, cortical thickness, white matter integrity, and cerebral blood flow in AD patients can be gleaned. Notably, the advent of deep learning techniques has emerged as a formidable approach for medical image analysis across a wide range of conditions, encompassing neurodegenerative disorders, orthopaedic ailments, and cancer. Leveraging the intrinsic capacity to automatically learn and extract intricate features from image data through the construction of multilayer neural networks, deep learning methods transcend conventional machine learning approaches by iteratively optimizing models with large-scale data. This obviates the need for manual feature engineering, engenders enhanced diagnostic accuracy, and improves overall diagnostic efficiency.

The integration of deep learning techniques into AD diagnostic research has been instrumental in the development of algorithms aimed at supporting physicians in early diagnosis and prognosis prediction. By harnessing the power of deep learning for MRI analysis, the detection of AD at its nascent stages becomes attainable, thereby augmenting the diagnostic capabilities and precision of healthcare professionals. This, in turn, facilitates the timely implementation of intervention strategies to mitigate further cognitive decline. The exploration of deep learning-based early AD diagnosis holds significant theoretical and practical implications, encompassing the identification of initial brain alterations in AD patients, the enhancement of AD diagnostic efficiency, the amelioration of the quality of life for individuals afflicted by AD, and the advancement of deep learning theory as a whole.

Deep learning has become increasingly prominent in medical image analysis, surpassing conventional machine learning algorithms in various domains (Lian et al., 2020). Notably, its automated feature learning capability from raw data, without the need for human intervention or domain expertise, distinguishes it as a highly advantageous approach. Among the array of deep learning models, convolutional neural networks (CNNs) have demonstrated remarkable success and widespread adoption for medical image analysis. This can be attributed to their proficiency in capturing both spatial and semantic information from images, thereby enabling robust and accurate analysis in the medical field.

Korolev et al. (2017) explored the use of 3D CNNs for AD classification and developed two 3D CNN models that achieved comparable results to traditional methods using ADNI data. Cheng et al. (2017) employed 3D CNNs for AD classification, but they amalgamated multiple 3D CNNs by training them on MRI data from distinct brain regions and subsequently appending an FC layer to each one. Improved AD diagnosis performance of 3D CNNs was achieved by Zhang et al. (2021) through the incorporation of an attention mechanism, which enabled the network to selectively focus on relevant features. Spasov et al. (2019) proposed a method to reduce the computational complexity of 3D CNNs by using separable convolution techniques.

Pan et al. (2022) invented an adaptive interpretable ensemble model (3DCNN+EL+GA) that leverages the power of 3DCNN, ensemble learning and genetic algorithm (GA) for AD classification and biomarker discovery. A total of 246 base classifiers (3DCNN) were trained on a dataset of 246 brain regions and a majority voting scheme was employed to select the optimal combination of base classifiers from the set of classifiers by using GA. Liu et al. (2020) developed a multi-task deep CNN model that performed both hippocampal segmentation and disease classification tasks simultaneously. They combined a 3D densely connected convolutional network (3D DenseNet) with the hippocampal segmentation results to learn richer features for AD diagnosis.

Khvostikov et al. (2018) built 3D CNNs for AD classification by extracting hippocampal ROIs from sMRI and DTI data. They also balanced the classes of different sizes by using data augmentation methods and investigated the effect of ROI size on classification results. Liu et al. (2018) developed an end-to-end approach for AD classification by extracting local fMRI image patches centred on predefined anatomical landmarks. These patches are applied to capture both the local and global structural features from the images. However, many of these 3D deep learning-based approaches still excessively rely on pre-determined ROIs before the training of the network, which may limit the performance due to the presence of irrelevant features in sMRI for AD diagnosis. Moreover, most of these studies only focus on binary classification, which is not very helpful for determining the stage of the patient’s situation.

Each individual’s brain exhibits unique characteristics and may possess disease-related features that cannot be fully captured by a single MRI slice. Consequently, the MRI slices of patients may exhibit minimal deviation from those of healthy individuals, making classification challenging. Previous studies have primarily focused on extracting features from specific slices or the entire image, disregarding the features of slices from different views and the complementary nature of features across these slices. Moreover, they have not effectively utilized the comprehensive structural information available in whole-brain MRI scans (Lian et al., 2022). In light of this, Qiao, Chen & Zhu (2021) proposed a novel approach for early AD diagnosis based on MRI, which involves extracting fused global features from multi-view slice features. They employed a simple splicing technique to combine the features of multiple slices from the same view. However, it is restricted by an assumption that all slices are of equal importance for the classification task. Different slices may contribute differently to disease features. Therefore, using equal weights for feature fusion may not effectively capture the relevant features, potentially resulting in lower classification accuracy.

Actually, the slices from the same view capture diverse brain regions and exhibit distinct features, essentially representing channel-specific mappings that reflect varying degrees of importance in the slice clusters. Additionally, it is essential to recognize that slices from different locations are not isolated entities but interconnected, collectively forming a comprehensive feature representation of the slice cluster in that specific direction. Hence, when fusing the features of different slices from the same view, it becomes crucial to consider both the significance of the information carried by each slice and the contextual relationship that exists between them. By incorporating these factors into the fusion process, a more robust and informative representation can be achieved, facilitating improved accuracy in capturing essential features for classification tasks, such as early AD diagnosis based on MRI data.

In this dissertation, a novel approach to the early diagnosis of AD called ASMF is proposed. Firstly, the Multi-view Slice-level Feature Extraction (MSFE) method is employed to acquire slice-level features from three distinct views (sagittal, coronal, and cross-sectional) by repeatedly slicing the 3D MRI and leverage three separate 2D sub-networks to extract features from each view. Then, an attention mechanism is incorporated to guide the fusion of slice features for each view, assigning varying weights to individual slices based on their respective importance. Secondly, global features are extracted from the entire MRI images by using a 3D CNN to complement the slice-level features. Finally, the slice-level and global features are fused to generate a more comprehensive feature representation for the classifier. The key contributions of this study can be concluded as follows: 1) To address the limitations of relying on feature extraction in a specific direction, this study proposes a novel MSFE-based approach for early diagnosis of AD.

2) This is the first study that incorporates a self-attention mechanism and the fusion of multi-view and multimodal features to construct a comprehensive representation of the MRI images for classification.

3) According to the experimental results, the proposed method outperforms other recently published promising approaches.

The remainder of this article is arranged as follows. The materials and methods are described in “Materials and Methods”. “Results and Discussion” provides experimental results and corresponding discussion. Finally, the summary of this study is given in “Conclusions”.

Materials and Methods

In contrast to previous studies that typically rely on a single slice from a specific view for AD diagnosis, we utilize multiple slices from three views of 3D MRI scans to extract features. As shown in Fig. 1, the workflow of ASMF contains the following steps: (1) three views of slices are separately processed by the proposed Slice Feature Extraction Network (SFEN) and Slices Fusion Attention (SFA) module to generate slice-level features; (2) the preprocessed 3D MRI volumes are sent to designed 3D neural network to produce volume-level feature representations; (3) the slice-level and volume-level features are merged in FC layer; (4) the classifier is trained to predict the correct category of AD, MCI and NC.

Figure 1 The framework of AMSF.

Figure source credit: ADNI.

Multi-view slicing feature extraction

To facilitate the analysis of the 3D MRI data, we initially partitioned it into three distinct planes: sagittal, coronal, and transverse planes. Each plane represents a different orientation of the brain and provides unique information about its structural characteristics. To ensure a comprehensive assessment, we extracted a total of 40 slices per view, thereby constructing a robust slice cluster. Figures 2 and 3 visually illustrate the 3D MRI images, showcasing the diversity and coverage achieved across the different planes. By encompassing multiple slices from each view, our approach captures a broader range of relevant features, enabling a more thorough examination of the brain’s structural attributes.

Figure 2 3D MRI data slicing in three directions.

Figure source credit: ADNI.

Figure 3 The examples of ADNI-I dataset.

(A) AD, (B) MCI, (C) NC. Figure source credit: ADNI.

To extract features from each slice of the 3D MRI data, we used a slice-level feature extraction network that takes slice clusters as input. A slice cluster consists of 40 slices from one of three possible views: sagittal (x), coronal (y) or transverse (z), which represent different orientations of the brain structure and contain different types of features. Therefore, we designed a separate SFEN for each view. As can be seen in Fig. 4, it consists of four blocks (including a 3*3 convolutional layer, BN layer with ReLU activation function and 2*2 max pooling layer) and 1*1 average pooling layer. For instance, the sagittal view (x), Cx denotes the cluster of slices in this direction. Each slice in this cluster has an index i that ranges from 1 to 40. Thus, Cx can be written as

Figure 4 The architecture of SFE.

Figure source credit: ADNI.

(1) Cx=[Cx1,Cx2,...,Cxn]

where n denotes the number of slices in the x-direction. The feature extraction of the i-th slice in the x-direction can be expressed as

(2) Txi=AvgPool(Fx(Cxi,Wxi))

where Fx represents the feature extraction function consisting of multiple blocks containing a 3*3 convolutional layer, BN layer, activation function ReLU and maximum pooling layer, as shown in Fig. 4. Besides, Wxi denotes the convolutional layer weight of the i-th slice in the x-direction, Tx stands for the features of the cluster of slices in the x-direction after Fx. Then the slice cluster feature in the x-direction can be expressed as

(3) Tx=[Tx1,Tx2,...,Txn]

Attention-based slice feature fusion

To integrate the distinct features within each view (x, y, or z), we leverage the notion of a “slice cluster” comprising 40 slices that possess unique characteristics. These features serve as mappings for specific brain structures observed from their respective perspectives. Notably, these features are not isolated entities, but rather interconnected across various locations within the slice cluster. Consequently, they collectively establish a comprehensive feature representation for the given view. To effectively merge these features, it is imperative to consider both their significance and their interdependencies. To address this, we propose a novel mechanism termed SFA. It employs self-attention to capture contextual information among the slices and assigns attention weights to each slice based on its relative importance and contribution to the overall feature representation of the view. By incorporating this attention-based weighting scheme, SFA effectively balances the significance of different slices while enriching them with contextual information derived from their interrelationships. Figure 5 provides an illustrative depiction of the structural composition of SFA.

Figure 5 The architecture of SFA.

We feed the slice cluster feature Tx into SFA as input. Tx has a dimension of 40*1*128 and it is obtained by concatenating the features of 40 slices along the channel dimension after applying SFEN. To reduce the number of channels from 128 to 1, we use a 1*1 convolution layer that compresses Tx into a single-channel feature map. This gives us an aggregated feature that represents the fusion of 40 slices. Next, we apply a Softmax function to Tx and multiply it with Sx. This way, we obtain Sx that contains contextual information between slices weighted by their attention scores. We can write this process as

(4) Sx=Tx∗∗Softmax(Conv(Tx))

where Conv means 1*1 convolution operation, and Tx∗ is obtained by reducing one dimension of Tx.

Sx contains the contextual relationship between different slices, which needs to be assigned to each slice by calculating the slice feature weights of different channels. Firstly, Sx is expanded by one dimension and the number of channels is reduced by 1*1 convolution, then BatchNorm and ReLU activation function operations are performed, and the number of channels is raised to the original number by 1*1 convolution, denoted as

(5) Ax=Conv(BR(Conv(Sx′)))

where Sx′ is obtained by adding one dimension to Sx, and BR denotes the BatchNorm and ReLU activation functions.

Ax is applied to represent the reweighted channel features that reflect how much each slice contributes to Sx. To obtain the slice fusion feature that incorporates both channel weights and contextual relationships between slices, we multiply Ax with Tx along the channel axis and sum them up. We denote this final output as

(6) Fx=∑i=140⁡Txi∗Axi

where 40 denotes the number of slices and Axi denotes the i-th channel of weight Ax.

Global feature extraction

The global feature extraction (GFE) component contains four blocks, each encompassing a sequence of operations: a 3D convolutional layer, a 3D batch normalization layer (BN), a rectified linear unit (ReLU) activation function, and a 3D maximum pooling layer. Subsequently, a 3D average pooling layer is employed to convert the multichannel features into a vector that encapsulates the global information. Figure 6 provides the structure of GFE. Following the acquisition of the global features, adaptive averaging pooling is applied to generate a one-dimensional vector. The multi-view slice-level features are then concatenated with the global vector. Finally, a fully connected layer is employed to obtain the ultimate classification outcomes.

Figure 6 The structure of GFE network.

Results and discussion

Dataset and processing

The Alzheimer’s Disease Neuroimaging Initiative (ADNI) dataset (http://adni.loni.usc.edu/) is employed in this study. ADNI provides data processed by standard volumetric analysis methods, including gradient non-linearity correction, B1 correction, N3 correction, CAT12 for extraneous tissue removal, alignment and smoothing operations. In this study, the ADNI-I with a total of 351 3D-MRI scans for NC subjects, 301 3D-MRI scans for AD subjects and 331 3D-MRI scans for MCI subjects are employed.

In the experiments, the ADNI-I data were utilized and underwent several preprocessing steps before being fed into the feature extraction network, as depicted in Fig. 5. Initially, the background information that is unrelated to the classification task was eliminated. Subsequently, the image size was adjusted to 90*90*90, and the image density was normalized based on the mean and standard values of the non-zero region. The 3D MRI data were then sliced according to three directions, and for each view, the middle 40 slices were selected on the basis of a preliminary experiment. Finally, feature extraction operations were conducted on the obtained slices according to the cross-validation protocol with 70% samples for training and the left 30% for testing.

Ablation experiments

To investigate the performance of the proposed framework, the indicators of Acc (Accuracy), Sen (Sensitivity), Spe (Specificity), Pre (Precision) and F1 (F1 score) are employed for evaluation. The batch-size for model training is 12, the number of epochs is set to 100, and the learning rate is 0.01. The ablation experiments are conducted with the detailed definition as follows: 1) 3D: 3D features are only obtained by 3D GFE.

2) 2D: only employ 2D features learned by MSFE.

3) 2D+3D: 2D and 3D features are extracted by MSFE and GFE.

4) SE+2D+3D: the combination of 2D+3D with SE.

5) AMSF (SFA+2D+3D): SFA-guided 2D+3D.

The results of the ablation experiments are shown in Table 1. It can be seen that the classification accuracy of 3D is the lowest (76.9%) as well as Sen, Spe, Pre and F1. The 2D method provides better performance with an improvement of 89.4%, which is 12.5% higher than 3D. With the help of fused global features, 2D+3D achieves 91.6% Acc with the second-ranking Spe (95.9%) and Sen (91.6%), which surpasses only using 2D or 3D strategy. However, with the combination of SE, SE+2D+3D cannot improve the performance further lagging behind in all indicators, especially for Sen with a 2% descent. It can be suggested that SE fails to effectively address the imbalanced importance among different slices within the same view for this particular task. At last, it is demonstrated that AMSF reaches the highest accuracy of 94.3%, surpassing that of the 2D+3D and SE+2D+3D by 2.7% and 4.5% respectively. It also exceeds in F1, Sen, Spe and Pre. This indicates that SFA effectively integrates contextual relationships between different slices, enabling it to balance their importance within a given view.

Table 1 Results of ablation experiments.

Methods	Acc (%)	Sen (%)	Spe (%)	Pre (%)	F1 (%)	
3D	76.9	76.6	88.7	77.2	76.7	
2D	89.4	90.2	94.8	90.8	89.8	
2D+3D	91.6	91.6	95.9	92.2	91.3	
SE+2D+3D	89.8	89.6	95.1	89.6	89.5	
AMSF	94.3	94.2	97.1	94.2	94.1	

The curve of validation loss for each epoch is shown in Fig. 7. It can be seen that during the training process, the training loss decreases continuously in the first 20 epochs and remains stable. The validation loss declines in the first 10 epochs, then fall in fluctuation before 25 epochs and finally stabilises after 35 epochs. Figure 8 gives the curve of the accuracy for each epoch. With a zigzag rise before 35 epochs, the validation accuracy reaches saturation. The confusion matrix can be seen in Fig. 9. The classification of AD achieves the highest accuracy (97.0%), indicating that ASMF is more sensitive to the features of AD. The classification accuracy for NC is 95.4% with 4.6% misclassifying NC to MCI. For MCI, it obtains the worst performance (90.4%), as well as an 8.2% misclassifying MCI to AD. It can be observed that the subtle differences in features between MCI and AD are formidable to extract, which is still a significant problem in AD diagnosis.

Figure 7 The curve of validation loss of AMSF.

Figure 8 The curve of accuracy of AMSF.

Figure 9 Confusion matrix of of AMSF in ablation experiment.

Comparison with other methods

An experimental comparison with previous promising approaches is also arranged, including two traditional machine learning-based approaches (JMMP-LRR, Liner SVM) and three deep learning-based methods (DemNet, Automatic CL and AdaBoost). The abstracts of these works are listed as follows: 1) Automatic CL (Gracias & Silveira, 2022): curriculum learning is employed in the early diagnosis of AD based on a 3D CNN network, by incorporating task complexity, cognitive test scores, and ROI features, thereby enhancing the accurate classification of MCI.

2) Linear SVM (Yuan, Yao & Bu, 2022): Mr cortical and ApoE4 gene features are explored for AD classification and the optimal performance is achieved by an SVM classifier with higher sensitivity and specificity.

3) JMMP-LRR (Sheng et al., 2020): it aims to better alleviate the problem of small-sample, ul-tra-high-dimensional features, accompanied by stable AD classification accuracy.

4) DemNet (Billones et al., 2016): an improved 16-layer VGGNet architecture is proposed with SOTA (state-of-the-art) classification results of AD vs MCI vs NC.

5) AdaBoost (Buyrukoğlu, 2021): an ensemble learning method is designed for AD diagnosis with SOTA performance compared with different ensemble learning methods.

Table 2 shows the classification results on the ADNI dataset compared with other previous promising methods. It can be seen that ASMF achieves the best performance with 94.3% classification accuracy, which is 1.6–7.1% higher than other related works. However, the Spe of ASMF (91.7%) is not the best like Sen among these approaches. Surprisingly, only by using a small-scale dataset, traditional machine learning-based methods (Linear SVM and AdaBoost) surpass the other three deep learning-based approaches. These findings are also verified in the ROC curve shown in Fig. 10, indicating that ASMF owns the better prediction ability for AD. Especially for Linear SVM, except for Sen, it outperforms Automatic CL and DemNet on Acc and Spe without the help of DNNs, which demonstrates the superior performance of traditional machine learning methods on a small dataset (only 100 subjects’ data). These methods are suitable for clinical applications and other scenarios of small datasets, but could not obtain the same scores as the larger and deeper models on actual open scenarios despite the higher complexity and the limited interpretability. These results provide important insights into the better solution for AD early diagnosis not only depending on the large scale of the dataset but also the organic combination of feature construction strategy and deep neural networks.

Table 2 The comparison results with other previous methods for AD vs MCI vs NC classification.

Methods	Category with the number of samples	Acc (%)	Sen (%)	Spe (%)	
Automatic CL	AD: 95, MCI: 207, NC: 104	87.2	86.5	87.8	
JMMP-LRR	AD: 24, MCI: 24, NC: 24	89.0	88.5	88.2	
DemNet	AD: 300, MCI: 300, NC: 300	91.9	92.4	91.3	
Linear SVM	AD: 34, MCI: 45, NC: 21	92.0	90.8	92.5	
AdaBoost	AD: 85, MCI: 193, NC: 111	92.7	92.5	93.1	
AMSF (ours)	AD: 301, MCI: 331, NC: 351	94.3	92.6	91.7	

Figure 10 The ROC curve of the compared models.

Conclusions

This study set out to address the limitation of specific direction feature extraction on a single modal of fMRI data, thereby a novel AD diagnosis approach called AMSF is proposed. Specifically, it incorporates (1) the features extracted from different views of 2D slices, (2) the attention mechanism and (3) volume-based 3D features, which aims to investigate the effectiveness of attention-based global and local feature representation for accurate diagnosis of AD. The experiments are conducted on the public ADNI-I dataset, which demonstrates our proposed method outperforms several previous approaches with 1.6–7.1% improvements. The study contributes to our understanding of the differences between traditional machine learning and deep learning methods on AD classification tasks. The insights may be of assistance to design appropriate models coping with various scales of the fMRI dataset for clinical applications. Although promising experimental results are achieved, the drawback still exists to be addressed. For example, a dataset with a limited size may restrict the generalization of the proposed model for practical applications. In future, we would try to explore (1) transfer learning, domain adaptation or domain generalization approaches, (2) multi-scale feature extractor on 2D slices and 3D volumes, and (3) other modalities of the dataset (such as PET, EEG or MEG) to further improve the ability for AD diagnosis.

Supplemental Information

Supplemental Information 1 The first part of Raw data from ADNI-I dataset applied for data analysis.

Click here for additional data file.

Supplemental Information 2 The second part of Raw data from ADNI-I dataset applied for data analysis.

Click here for additional data file.

Supplemental Information 3 The implementation code of AMSF.

Click here for additional data file.

Data used in preparation of this article were obtained from the Alzheimer’s Disease Neuroimaging Initiative (ADNI) database (adni.loni.usc.edu). As such, the investigators within the ADNI contributed to the design and implementation of ADNI and/or provided data but did not participate in analysis or writing of this report. A complete listing of ADNI investigators can be found at: http://adni.loni.usc.edu/wp-content/uploads/how_to_apply/ADNI_Acknowledgement_List.pdf.

Additional Information and Declarations

Competing Interests

Author Contributions

Data Availability

The authors declare that they have no competing interests.

Yameng Zhang conceived and designed the experiments, performed the experiments, analyzed the data, performed the computation work, prepared figures and/or tables, authored or reviewed drafts of the article, and approved the final draft.

Shaokang Peng conceived and designed the experiments, performed the experiments, analyzed the data, performed the computation work, prepared figures and/or tables, authored or reviewed drafts of the article, and approved the final draft.

Zhihua Xue performed the experiments, analyzed the data, performed the computation work, prepared figures and/or tables, and approved the final draft.

Guohua Zhao performed the experiments, analyzed the data, performed the computation work, prepared figures and/or tables, and approved the final draft.

Qing Li performed the experiments, analyzed the data, performed the computation work, prepared figures and/or tables, and approved the final draft.

Zhiyuan Zhu performed the experiments, analyzed the data, performed the computation work, prepared figures and/or tables, and approved the final draft.

Yufei Gao conceived and designed the experiments, authored or reviewed drafts of the article, and approved the final draft.

Lingfei Kong conceived and designed the experiments, authored or reviewed drafts of the article, and approved the final draft.

The following information was supplied regarding data availability:

The data is available at Kaggle: https://www.kaggle.com/datasets/tourist55/alzheimers-dataset-4-class-of-images/data.

The data used in the preparation of this article are available from the Alzheimer’s Disease Neuroimaging Initiative (ADNI) database: https://adni.loni.usc.edu.

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
