# Peer review of "AMSF: attention-based multi-view slice fusion for early diagnosis of Alzheimer’s disease"

_PeerJ Computer Science, doi:10.7717/peerj-cs.1706_

## Round 0.1 · original submission · Major Revisions

Dear authors,

Thank you for your submission. Your article has not been recommended for publication in its current form. However, we do encourage you to address the concerns and criticisms of the reviewers and resubmit your article once you have updated it accordingly. Reviewer 2 has requested that you cite specific references. You may add them if you believe they are especially relevant. However, I do not expect you to include these citations, and if you do not include them, this will not influence my decision.

Best wishes,

**Language Note:** The review process has identified that the English language must be improved. PeerJ can provide language editing services - please contact us at copyediting@peerj.com for pricing (be sure to provide your manuscript number and title). Alternatively, you should make your own arrangements to improve the language quality and provide details in your response letter. – PeerJ Staff

Reviewer 1 ·

Basic reporting

In the study, researchers proposed an attention-based multi-view slice fusion method for the early diagnosis of Alzheimer's disease. Alzheimer's disease is a disease that exists today and whose treatment is not clear. Therefore, the problem is important and its solution is necessary. Although this study is different from the studies existing in the literature, making the revisions requested will increase the readability and quality of the study.

- In the Abstract, the results of the study should also be added and the Abstract should be digitized.

Experimental design

- A flow chart of the study is not given. A flow chart of the study should be given under the section of "Materials and Methods" and the chart should be briefly explained.

- No detailed information is given about the data set. The number of images in the data set, how the images used in the study were selected, and whether training-test data or cross-validation was used while training the model were not mentioned. In addition, at least 5 of the visual data used in the study should be added to the article.

Validity of the findings

- The results obtained in the study should be presented in tables and interpreted.
- Researchers determined the performance of the model in the classification with accuracy, precision, sensitivity, specificity and F1-score evaluation metrics. However, only the accuracy score was interpreted. What the F1-score, precision, specificity and sensitivity scores represent should be discussed and interpreted.
- AUC score is an important criterion in biomedical and health studies. It would be to the benefit of researchers to add this criterion to the article.
- How many epochs was the model trained with in the study? If it is trained with 35 epochs, this epoch value is very low. Why was the model trained with a low epoch?
- The confusion matrix was not included in the study. The confusion matrix should be added and the results interpreted.
- The results obtained in the study were compared with other studies. This comparison should be made in a table and the results obtained in those studies should be added to the table and the results in this study should be interpreted.
- The advantages and disadvantages of this study should be mentioned and the performance of the model should be discussed.

Reviewer 2 ·

Basic reporting

The study proposes a novel approach in the diagnosis of Alzheimer's disease (AD) utilizing attention-based multi-view slice fusion (AMSF) combined with traditional and deep learning methods. While the effort to use multiple slices and views for an enhanced understanding of the brain's state is commendable, several aspects of the paper require critical assessment.
• The use of attention mechanisms in neural networks has been well-documented, and the study's claim that this is the “first study that incorporates the self-attention mechanism” for MRI images in AD diagnosis (lines 174-175) seems overreaching. References to prior related works using attention mechanisms in similar settings are missing. The authors are invited to check and discuss Ramya et al. Alzheimer’s Disease Segmentation and Classification on MRI Brain Images Using Enhanced Expectation Maximization Adaptive Histogram (EEM-AH) and Machine Learning (2022) Information Technology and Control, 51 (4), 786 – 800. Odusami et al. Pixel-Level Fusion Approach with Vision Transformer for Early Detection of Alzheimer’s Disease (2023) Electronics, 12 (5), 1218. Misra et al. Explainable Deep-Learning-Based Diagnosis of Alzheimer’s Disease Using Multimodal Input Fusion of PET and MRI Images (2023) Journal of Medical and Biological Engineering, 43 (3), 291 – 302. Maskeliūnas et al. Pareto Optimized Adaptive Learning with Transposed Convolution for Image Fusion Alzheimer’s Disease Classification (2023) Brain Sciences, 13 (7), 1045.
• The success of deep learning models largely depends on the dataset's size. The study hints at the use of a small dataset (line 338-339). This raises concerns about the model overfitting and its generalizability to larger, diverse populations. It is of vital importance to address the potential overfitting in a more rigorous manner.
• It's intriguing that traditional machine learning-based methods, especially the Linear SVM, outperformed some deep learning-based approaches, even without the aid of DNNs (line 339-340). This should have been highlighted and explored more. What does this imply about the complexity and interpretability of deep learning methods in this context?

Experimental design

• The paper lacks clarity in places. For instance, the distinction between 'slice-level' and 'global' features is not explicitly explained, which might lead to confusion. Diagrams or illustrative figures would have been beneficial here.
• The design and specifics of the Slice Feature Extraction Network (SFEN) for each view (line 206) are scarcely detailed. It would be pivotal for replicability to have more information on the architecture, training procedures, and hyperparameters.
• Extracting 40 slices per view might introduce redundancy. The rationale for selecting 40 slices and the potential implications of feature redundancy should be critically examined.
• While the classification accuracy is provided (94.3%), other metrics like sensitivity, specificity, or the area under the ROC curve, which are essential in medical diagnosis scenarios, are not mentioned. These metrics would provide a more comprehensive evaluation.
• In medical applications, the interpretability of models is of utmost importance. The paper doesn't touch upon how clinicians might interpret or trust the model's predictions, given its complex nature.

Validity of the findings

• The conclusion section (line 346-356) seems repetitive, echoing the abstract. A more in-depth reflection on the results, comparison with prior studies, and potential applications would enhance the value of this section.
• While the study acknowledges the need for larger datasets (line 355-356), it would be beneficial to discuss other potential improvements or extensions. For instance, are there other views or modalities that might be incorporated? Could transfer learning play a role in mitigating the dataset limitation?

Additional comments

This paper presents an interesting approach to Alzheimer's diagnosis; however, to enhance its quality, the authors should:
• Revisit the claims of novelty and provide better contextualization with existing literature.
• Address the concerns related to the dataset size and potential overfitting directly.
• Offer a deeper exploration of the performance of traditional machine learning methods compared to deep learning approaches.
• Improve clarity by providing diagrams, expanding on technical details, and enhancing the structure of the paper.
• In its current form, the article provides a foundation, but more rigorous work and clarity are required for it to make a significant impact in the field.

---

## Round 0.2 · accepted · Accept

Dear authors,

Thank you for the revision. In the opinion of the reviewers the paper is improved. The paper is now ready to be accepted.

Best wishes,

Reviewer 1 ·

Basic reporting

No comment

Experimental design

No comment

Validity of the findings

No comment

Additional comments

The researchers made the requested revisions. Therefore, the article can be published in its current form.

Reviewer 2 ·

Basic reporting

The authors have addressed all the requests and comments well. The quality of the manuscript is good. The manuscript can be accepted for publication.

Experimental design

The research was performed correctly.

Validity of the findings

The findings of the manuscript are valid.

Additional comments

I have no further comments.